# National Radon Action Plans in Europe and Need of Effectiveness Indicators: An Overview of HERCA Activities

**DOI:** 10.3390/ijerph19074114

**Published:** 2022-03-30

**Authors:** Francesco Bochicchio, David Fenton, Heloísa Fonseca, Marta García-Talavera, Pierrick Jaunet, Stephanie Long, Bård Olsen, Jelena Mrdakovic Popic, Wolfgang Ringer

**Affiliations:** 1National Center for Radiation Protection and Computational Physics, Italian National Institute of Health (ISS—Istituto Superiore di Sanità), 00161 Rome, Italy; 2Office of Radiation Protection and Environmental Monitoring, Environmental Protection Agency (EPA), Dublin 14, Ireland; d.fenton@epa.ie (D.F.); s.long@epa.ie (S.L.); 3Emergency and Radiation Protection Department, Portuguese Environmental Agency (APA—Agência Portuguesa do Ambiente), 2610-124 Amadora, Portugal; heloisa.fonseca@apambiente.pt; 4Nuclear Safety Council (CSN—Consejo de Seguridad Nuclear), 28040 Madrid, Spain; mgtm@csn.es; 5Ionizing Radiation and Health Department, French Nuclear Safety Authority (ASN—Autorité de Sûreté Nucléaire), 92120 Montrouge, France; pierrick.jaunet@asn.fr; 6Norwegian Radiation and Nuclear Safety Authority (DSA), 1361 Østerås, Norway; bard.olsen@dsa.no (B.O.); jelena.popic@dsa.no (J.M.P.); 7Department for Radon and Radioecology, Austrian Agency for Health and Food Safety (AGES), 4020 Linz, Austria; wolfgang.ringer@ages.at

**Keywords:** radon, policy, regulations, indicators, effectiveness, national radon action plans, radon programs, radioprotection

## Abstract

Protection of the population and of workers from exposure to radon is a unique challenge in radiation protection. Many coordinated actions and a variety of expertise are needed. Initially, a National Radon Action Plan (NRAP) has been developed and implemented by some countries, while it is currently recommended by international organizations (e.g., World Health Organization) and required by international regulations, such as the European Council Directive 2013/59/Euratom and the International Basic Safety Standards on Radiation Protection and Safety of Radiation Sources, cosponsored by eight international organizations. Within this framework, the Heads of the European Radiological Protection Competent Authorities (HERCA) have organized activities aimed at sharing experiences to contribute toward the development and implementation of effective NRAPs. Two workshops were held in 2014 and 2015, the latter on radon in workplaces. As a follow-up to these, an online event took place in March 2021, and a second specific workshop on NRAP is planned for June 2022. These workshops were attended by experts from the competent authorities of European countries, relevant national and international organizations. The experience of several countries and the outcomes from these workshops have highlighted the need for adequate indicators of the effectiveness and progress of the actions of NRAPs, which could also be useful to implement the principle of optimization and the graded approach in NRAPs. In this paper, the activities of HERCA to support the development and implementation of effective NRAPs are described and some examples of effectiveness indicators are reported, including those already included in the NRAP of some European countries.

## 1. Introduction

After smoking, exposure to radon is a major cause of lung cancer worldwide, and it is one of the leading causes among never-smokers. In Europe, it is deemed to be responsible for about 20,000 cancer deaths per year [1]. Accordingly, protection against radon is one of the actions included in Europe’s Beating Cancer Plan [2].

Protection of population and workers from exposure to radon is a unique challenge in radiation protection as it is an exposure that takes place in nearly all indoor environments. Moreover, reducing population exposure to radon and related health risks requires action across various areas, such as building, occupational health or technical service provision. It also requires, not only regulation and political will from national, regional and local governments, but also public awareness, as well as the drive and means for individuals to take action at a reasonable cost when needed. This can only be achieved by means of a national policy and strategy, usually referred to as the “National Radon Action Plan” (NRAP).

Therefore, some countries, including European ones, after having carried out a national survey and other activities to evaluate the national situation and developed experience in remedial actions and other relevant services, set up and implemented NRAPs, such as Switzerland (since 1994), Czech Republic (since 2000) and Italy (since 2002). On the basis of these national experiences, the requirement of developing NRAPs has also been included in international recommendations and regulations. In fact, the World Health Organization, in 2009, recommended “national radon programs to reduce both the risk for the overall population exposed to an average radon concentration and the risk of individuals living with high radon concentrations” and presented “components for developing a national radon programme and a framework for the organization of such a programme at the country level” [3]. Moreover, in 2014, the International Commission on Radiological Protection (ICRP), in its recommendation on protection against radon exposure, reported that “the national radon protection strategy should be implemented through a national radon action plan established by national authorities with the involvement of relevant stakeholders” and that “the strategy should have a commitment to reduce the overall exposure of the general population and the highest individual exposures” [4]. Building upon these, in 2014, the European directive 2013/59/Euratom established the requirement for the Member States to develop and implement national radon action plans, with the ultimate goal to reduce cancer deaths in their territories [5]. A similar requirement has been introduced in the International Basic Safety Standards on Radiation Protection and Safety of Radiation Sources, cosponsored by eight international organizations (EC, FAO, IAEA, ILO, OECD/NEA, PAHO, UNEP, WHO) [6].

To support countries in developing their NRAPs, the Heads of European Radiological Competent Authorities (HERCA) organized a first specific workshop in 2014. Other related activities were carried out in subsequent years, including the organization of a workshop in 2015 focused on radon in workplaces that also covered NRAPs, the creation of the HERCA Working Group in 2017 on “Natural Radiation Sources” (WG NAT), and organization of an online pre-workshop event in March 2021 as an introduction to the second HERCA workshop focused on NRAPs that will be held in June 2022.

The European picture regarding the measures implemented by European countries against radon exposure has drastically changed in this eight-year time interval between the two workshops. Whereas very few countries had set into action an NRAP in 2014, most of them have already done so and now have regulations governing radon in workplaces and dwellings. Thus, the focus of the Second HERCA Workshop on NRAPs in 2022 will be on the effectiveness of different actions implemented through those NRAPs as well as on the selection of proper indicators to monitor progress towards the goal to reduce radon-related cancers in Europe.

In this paper, after a short description of the evolution of protection against radon in Europe, including a summary of the current EU legal framework on protection from radon exposure and its transposition into national legislation of Member States (Section 2), the HERCA workshops and on-line event on NRAPs are described (Section 3, Section 4 and Section 5) and the issue of indicators is presented and discussed (Section 6), reporting both the HERCA on-going activities on this issue and examples from the NRAP of some European countries.

## 2. Evolution of the Legal Framework on Protection against Radon in Europe and NRAPs

The first evidence that radon could cause lung cancer dates back to the early 20th century, based on the high prevalence of the disease in miners from Czechoslovakia [7,8]. Using radon measurements conducted in Czechoslovakia uranium mines since 1949, a study on lung cancer risk, initiated in the late 1960s, was the second one worldwide (after the Colorado Plateau study) to establish the risk related to cumulated exposure to radon progeny [9]. From the 1970s on, radiation protection measures for uranium miners, including radon monitoring, started to be implemented in several European countries.

It was not until the mid-1980s that exposure to radon was recognized not only as an occupational hazard, but also as a potential public health problem, because of the occurrence of elevated radon levels in dwellings, such as in Sweden [10].

In Europe, protection from exposure to radon in dwellings was subject of an EU Commission Recommendation of 21 February 1990 [11]—not binding to EU Member States—which recommends, for existing dwellings, to consider simple and effective measures aimed at reducing radon levels where these exceed the value of 400 Bq/m^3^, whereas, for future constructions, a design level of 200 Bq/m^3^ was recommended. As regards to work activities, in 1996, the Directive 96/29/Euratom [12] included, in Title VII, some general requirements for the Members States to control exposure to natural radiation sources (including exposure to radon), in particular for work activities (unrelated to the nuclear fuel-cycle) involving enhanced levels of exposures, such as mines, spas, caves, underground workplaces and aboveground workplaces in identified areas. One year later, the European Commission published recommendations for the implementation of Title VII of the Directive 96/29/Euratom [13]. Both 1990 recommendations on radon in dwellings and Directive 96/29/Euratom were replaced by the Directive 2013/59/Euratom.

The current EU legal framework on radon protection is laid out in the Directive 2013/59/Euratom (often, and hereafter, referred to as the EU-BSS directive or EU-BSS). This directive is binding to the EU Member States, but it has also been adopted by a number of countries outside the EU, including Norway, UK and Switzerland. A new requirement of the EU-BSS with regard to radon protection is the obligation of the Member States to develop national radon action plans “addressing long-term risks from radon exposures in dwellings, buildings with public access and workplaces for any source of radon ingress, whether from soil, building materials or water” (article 103.1). This article further requires countries “to ensure that appropriate measures are in place to prevent radon ingress into new buildings” (art. 103.2) as well as to identify radon priority areas, defined as “areas where the radon concentration in a significant number of buildings is expected to exceed the relevant national reference level” (art. 103.3). Items to be considered in the NRAP are reported in the Annex XVIII of the EU-BSS.

Moreover, radon exposure in workplaces is specifically addressed in articles 54 and 35.2, whereas radon exposure in dwellings is specifically addressed in article 74. Reference levels not exceeding 300 Bq/m^3^ shall be established for both workplaces and dwellings. Workplaces where, despite remedial action, annual radon concentrations remain above the reference level are subject to notification and exposure of workers receiving annual doses above 6 mSv/y (or to the corresponding integrated radon exposure) shall be managed as a planned exposure situation. The limit of 20 mSv/y for occupational planned exposures to ionizing radiation also apply to radon exposure. Adopting the most recent ICRP dose coefficients [14], 6 and 20 mSv/y correspond to about 450 and 1500 Bq/m^3^, respectively, for standard occupancy of 2000 h/y.

HERCA WG NAT carried out recently a survey to explore how European countries had progressively adopted EU-BSS into their national legislation. Results from 20 HERCA member countries (shown in Figure 1), with regard to regulation on radon in workplaces and new buildings, are given in Figure 2. In particular, as per the survey, the federal Government of Slovakia (then integrated in Czechoslovakia) was a pioneer in 1972 in introducing regulation on radon in workplaces, followed in 1985 by the UK. On the other hand, in 1980, Sweden was the first European country to introduce compulsory limits for existing and new dwellings.

Moreover, as of February 2022, 90% of the countries surveyed have an NRAP in place. The evolution showing when countries established their first NRAP is displayed in Figure 3 in relation to the dates for approval and transposition deadline for EU-BSS directive, as well as with relevant HERCA events. NRAPs are generally reviewed and updated periodically, with time periods varying from 1 to 10 years.

Although there is a number of common items to be considered when developing an NRAP (as indicated in Annex XVIII of the EU-BSS directive), the plan needs to be adapted to the prevailing national circumstances. The radon problem widely varies among countries. Radon levels in dwellings are related to soil characteristics, as well as to a number of other factors, including building characteristics of the housing stock, climate or habits. Therefore, the national average radon level varies among countries, and this is, in turn, reflected in the number of cancer attributable to radon in the different countries [15,16,17,18,19]. For example, in the Netherlands, a portion of about 4% of lung cancers may be attributable to radon exposure, while in Sweden, the percentage rises to about 20% [20].

On the other hand, with radon exposure being a public health problem, the resources and efforts dedicated to the NRAP need to be commensurate to other health-related policies in the country.

Lastly, it needs to be acknowledged that whereas most countries have adopted reference levels of 300 Bq/m^3^, most radon attributable cancers occur at concentrations below 100 Bq/m^3^ (e.g., [21]). Consequently, although such reference levels are effective in mitigating the risk for the most exposed individuals, emphasis on optimization below the reference level is also needed. This is also in line with the principle of optimization and the graded approach for radioprotection introduced by ICRP in Publ.103, which recommended “reference level” (RL) to be used as a tool towards optimization, replacing the previous “action level” (AL) [22]. This implies that optimization has to be performed with priority for radon concentration levels above the reference level (RL), but optimization shall continue to be implemented for levels lower than the RL, whereas the previous AL tool required to consider remedial actions only for radon concentration levels above AL [4,21]. Reference level, as well as graded approach, has been largely applied by ICRP in its recommendation on protection from radon and also adopted by the Directive 2013/59/Euratom and the International BSS [5,6].

## 3. First HERCA Workshop on NRAP (2014)

In 2014, the same year the EU-BSS directive was published, the first HERCA workshop on NRAP was held [23]. The workshop was initiated in response to the EU-BSS directive [5] requirement for Member States to define and adopt an NRAP for reducing radon exposure. At that time, only a few European countries already had an NRAP, while most countries did not have or had just started planning for the development of such a plan (Figure 3). The major objectives of the workshop were to facilitate the preparation or updating of NRAP by jointly addressing steps and activities in the implementation of the requirements in the EU-BSS directive and providing a forum for European countries to exchange information, experience and challenges. The workshop was organized, on behalf of HERCA, jointly by the French Nuclear Safety Authority (ASN) and the Norwegian Radiation Protection Authority (NRPA, now Norwegian Radiation and Nuclear Safety Authority, DSA) and hosted by ASN in their premises in Paris from 30 September to 2 October 2014.

The World Health Organization (WHO), the International Atomic Energy Agency (IAEA) and the European Commission (EC) supported the workshop. In addition to these international organisations, representatives from radiation protection authorities and other relevant authorities from more than twenty countries were participating. About 90 participants attended the workshop. The workshop consisted of four separate working sessions in addition to an opening and a closing session. The working sessions raised issues on: (1) global strategy and NRAPs, (2) actions to reduce exposure in dwellings, (3) actions to reduce exposure in workplaces and buildings with public access, and (4) strategy for communications. Time for intensive discussions was included in each of the sessions.

The presentations from the international organisations generally focused on the main items of the global radon strategy. The WHO presented the WHO Handbook on Indoor Radon [3] and highlighted the need for national radon programmes, including multi-agency collaboration, the role of policy makers and authorities, financial considerations, mandatory versus voluntary approaches and establishing a national reference level. The IAEA representative presented the radon requirements in the International BSS [6], one such requirement being the implementation of an NRAP. The basic approach to developing an NRAP, comprising of information on radon levels in the country, radon measurements and radon national action plan as key issues, was considered. The representative from the European Commission reported on the legal basis, history, background, legal status and objectives of Euratom radiation protection legislation in general and of, at that time, the new EU-BSS directive in particular [5].

Speakers from European countries, Canada and US, emphasized the importance of national strategies for radon reduction, although somewhat different approaches for radon reduction were presented. The importance of intensive multi-level collaboration was highlighted, and it was considered as efficient to have one authority that coordinates the radon reduction activities and follows up the NRAP. However, certain differences regarding voluntary and mandatory approaches were noticed and discussed. It was concluded that the national radon programmes presented during the workshop had been successful, but that there still was a need to review certain aspects, especially to increase awareness and to speed up the pace of measurement and remediation measures. 

Furthermore, actions to reduce radon exposure in dwellings as well as workplaces and buildings with public access were discussed in more detail, including aspects in strategies for communication. Somewhat different national reference levels (100, 200 and 400 Bq/m^3^) were presented, and it was concluded that an update in relation to the requirement of the EU-BSS directive (reference level not greater than 300 Bq/m^3^) was needed in some countries. It was agreed that an NRAP should be based on knowledge, both from surveys of indoor radon concentrations in different buildings and from surveys of radon awareness. The identification and remediation of existing dwellings was identified as one of the biggest challenges related to radon exposure reduction.

Many countries had different experiences with a mandatory approach for radon, e.g., in the home buying/selling processes, for rental accommodation, for schools and kindergartens and for different types of workplaces. Advantages and disadvantages were discussed. One observation was that in order to comply with the EU-BSS directive, radon reference levels for workplaces needed to be redefined in many countries and that for some of them, this would pose a great challenge to regulatory systems. An overall agreement about the installation of preventive measures when constructing new homes was reached.

It was also agreed that radon risk communication should be an important aspect of an NRAP. Information on radon and related health risk should be given to different groups of people, including homeowners, landlords, employers, solicitors, estate agents, building professionals, architects, radon remediators, officers in local and national government and family doctors. Different communication channels should be considered to reach different groups of people. Experiences with positive effects of legislation in certain radon domains on the radon awareness in the general public and spin-off effect in other radon domains had been observed in some countries.

The closing session summarized the workshop by providing reports from each session, with EC giving their point of view on the main issues raised, and finally by presenting the main findings of the workshop. Based on overall presentations and discussions, it was concluded that radon is and should be confirmed as a public health issue and that the long-term goal of an NRAP is to reduce lung cancer risk was among the main conclusions. From the discussions during the workshop, it became clear that an NRAP should not only aim to reduce the high radon levels. It should also aim to reduce the average radon concentrations in the housing stock and other public access buildings and premises, thus reducing the overall lung cancer risk. Both voluntary and mandatory approaches should be considered for different types of buildings. An NRAP should also include a risk communication strategy to encourage people to utilize radon mitigation, focus on preventive measures, which is shown to be cost-effective, and that an imperative for success is the cooperation between different sectors at the national, regional and local level. The workshop also concluded that an NRAP should be evaluated and updated regularly. However, since most of the participating countries had not yet implemented an NRAP, this issue was not discussed on a large scale during the workshop.

## 4. The HERCA Workshop on Radon in Workplaces (2015)

The EU-BSS directive strengthens the requirements in terms of radon risk management in workplaces while, at the same time, leaving the Member States a degree of flexibility and the responsibility to determine some appropriate requirements: reference level, identification of types of workplaces where measurements are required, which requirements of a planned exposure situation must be applied where the exposure of workers is liable to exceed an effective dose of 6 mSv per year. 

In that context and on the basis of the first workshop’s main findings, FOPH (Switzerland), NRPA (Norway) and ASN (France), organized on behalf of HERCA a workshop to deepen EU-BSS requirements dedicated to radon in workplaces, including radon in buildings with public access and radon produced by materials in NORM activities [24]. The workshop was hosted by the International Labour Organisation (ILO), in Geneva, on 12–14 October 2015. The main objective was to provide a forum for European countries to exchange points of view on the transposition of the specific requirements of the EU-BSS dealing with radon exposure in workplaces and the impact on the existing national approach and strategy, in order to facilitate the transposition works.

Nineteen HERCA members and relevant international organizations such as the WHO, IAEA, ILO and ICRP attended the workshop.

After a session dedicated to both international and national exchange of experience, different viewpoints were shared on how to understand the specific requirements dealing with radon exposure in workplaces.

The issues raised were mainly focused on:The justification of actions to reduce radon exposures in the establishment and implementation of the National Radon Action Plan, being under the responsibilities of the Government and regulators;Identification of workplaces, radon measurements and control, the employer’s and/or the undertaking’s responsibilities depending on workplace type;The use of the reference level concept and how it supports, but it should not be confused with dose limitation.

A common understanding was proposed by HERCA members of the EU-BSS comprising 15 recommendations aimed at supporting the preparation or the updating of the National Radon Action Plans and associated regulations. These recommendations reaffirmed the importance of NRAPs, given the public health issues at stake, specified the objectives and the content of these Plans. While in general radon mitigation strategies in the NRAP may include both voluntary and mandatory approaches, HERCA supported the regulatory approach for radon measurement and mitigation in workplaces, including buildings with public access.

HERCA considered that the strategy used for reducing radon exposure in workplaces should be based on both preventive and guidance actions. It was underlined that the radon risk communication should be considered as a key aspect of any radon action plan and pointed out the necessity to involve and to inform or train, with targeted strategies, all stakeholders: employers, workers, their representatives (e.g., Labour Unions and Employers organization), occupational health services, bodies in charge of measurements and remedial actions, inspectors, etc.

The HERCA recommendations also highlighted the importance of making measurement results be representative of the annual average concentration in air. In the case where these measurements are mandatory, they should have to be carried out by bodies recognized by national authorities.

The EU-BSS directive requiring management of the radon risk by both the concentration of radon in air and the effective dose, the availability and the use of international guidelines, and associated tools, to calculate annual effective doses was identified as a key point. In case of doses exceeding 6 mSv per year, it was recommended to apply the occupational exposure requirements related to optimization, to the radiological surveillance of workplaces (adapted to radon exposure), to workers’ information and, in some cases, to individual monitoring.

Moreover, the necessity was emphasized to include in the NRAPs the assessment modalities of the radon regulatory requirement implementation, and of the collection and use of the radon measurements results in order to assess the impact of the strategies.

Lastly, to facilitate the implementation of the EU-BSS requirements on radon in workplaces, HERCA recommended the European Commission to provide European Guidance based on good practices. The publication of the document “Radon in workplaces—Implementing the requirements in Council Directive 2013/59/Euratom” in the Radiation Protection Series by the European Commission responds to that expectation [25].

## 5. Second HERCA Workshop on NRAPs (2022) and the Pre-Workshop Event (2021)

Following both the EU-BSS directive requirements and the conclusions from the first HERCA workshop on NRAP about regular review and update, HERCA WG NAT included in its action plan 2018–2021 an activity of organizing a second HERCA workshop specifically focused on NRAPs, as a follow-up of that one carried out in 2014. The main objectives of this second workshop are set as exploring the progress made in the implementation of NRAPs across Europe, analyzing the main issues regarding NRAP effectiveness indicators, international exchange on the best practices and lessons learned related to NRAP, including optimization and application of the graded approach to radon control at workplaces and the engagement and role of stakeholders in the implementation of NRAP. The follow-up NRAP workshop was planned for autumn 2020, but due to world pandemic conditions was postponed twice, and is currently planned for 21–23 June 2022 in Lisbon, Portugal.

However, to meet the significant international interest in NRAP issues, an international event, an introduction to the second HERCA NRAP workshop, was organized as an online event on 23 March 2021 [26]. The event was hosted by the Istituto Superiore di Sanità (Italian National Institute of Health).

The Programme of this event consisted of sessions that provided (1) an update on initiatives and supportive actions, with a special focus on NRAPs, from the international organizations (EC, IAEA and WHO), (2) insight into some core or new activities in national NRAPs, i.e., radon prevention in new buildings, regulatory framework for radon in workplaces, and citizen science in radon testing and remediation, and (3) a panel discussion with an international exchange on best practices, challenges and lessons learned.

Additionally, an online survey was organized for the participants at this online event. The survey included questions about:national radon reference level in dwellings and in workplaces;remediation rate in countries participants;existence of performance indicators to assess the impact of the NRAP;participation of different institutions and stakeholder engagements in NRAP activities;possible decline in the mean radon exposure due to the efforts to control radon and the implementation of the NRAP;information about awareness and use of different radon publications from EC, IAEA and WHO.

Responses were received from participants from eighteen European countries. It was shown that the reference levels for radon, both in dwellings and workplaces, is 300 Bq/m^3^ for the majority of responding countries, while only a very limited number of respondent countries had other reference values (Figure 4).

The remediation rate in different countries was shown to be below 20% in most countries, and only a few values were above 20%. Furthermore, it was confirmed that European countries have, according to the EU-BSS directive, implemented an NRAP, with the participation and collaboration of different authorities such as radiation protection, public health, work safety, accreditation, regional and municipal authorities. However, a significant decline in the mean radon exposure due to the efforts in the past decades to control radon and the implementation of the NRAP has been observed only in about 20% of the European countries with participants in this survey. Finally, the importance of performance indicators to assess the impact of the NRAP but also the need for future development and national implementations of these have been observed based on the survey results.

The analysis of the survey responses provided a kind of snapshot of the status of national radon control strategies and NRAPs, which will be further used in preparation for the second HERCA workshop on NRAPs in 2022.

Based on the presentations, the panel discussion and the survey results, key messages of this pre-workshop event were as follows:National Radon Action Plan is an important tool for ensuring the practical implementation of the diverse actions at the national level;Sharing experience on different topics of NRAPs are needed as countries are at different stages of implementation of activities;Radon prevention in new buildings, the approach to radon in workplaces and related risk communication as well as raising awareness and engagement level are examples of core issues where sharing of lessons learned, best practices and challenges is of high importance;Evaluation of taken measures is necessary for an assessment of their effectiveness, but also for further actions and the decision-making on recommended versus legally binding, compulsory approaches.

## 6. Effectiveness Indicators

One of the first proposals of possible useful indicators to evaluate a national program on radon can be found in the final WHO report of the ECOHEIS project, cosponsored by the European Commission-DG Sanco, regarding the development of environmental and health indicators for European Union countries [27]. Regarding radon exposure, in addition to the distribution of radon levels as an indicator of the exposure status, the following indicators of the progression of policy implementation were proposed: (i) the proportion of dwellings detected exceeding the national action level, (ii) the proportion of houses remediated among the ones with radon above action levels.

Appropriate indicators are needed to evaluate the effectiveness of the measures to be implemented in the NRAP. This is true in all countries. The main purpose of indicators is to ensure that implementation of the NRAP is resulting in progress towards the overall goal of the NRAP, i.e., to reduce the exposure of the population to radon and related health impact. Moreover, some indicators are also useful to evaluate the progress towards some specific goals that contribute to the overall goal. Indicators can also be used to guide the development of the NRAP and are also valuable to inform the evolution of the Plan over time. Such indicators could include, for example, the rate of radon testing among the public as well as the rate of radon mitigation among those having high radon concentrations. Moreover, indicators of baseline data could be useful, such as, for example, the national average indoor radon concentration, compiled or, where necessary, gathered at the start of the Plan and repeated at appropriate intervals through its lifetime. Additional indicators may need to be developed to assess the evolution of key factors influencing radon risk such as: location, age of construction, building type, etc.

It is worth noting that considerable differences exist between countries in the approaches used to address the radon risk. This is understandable given the diverse climates, geologies and building practices that exist between countries. In addition, while some countries have well developed NRAPs, that in some cases were published before the adoption of the EU-BSS directive in 2013, other countries have only recently published their NRAP and are at an early stage in defining the radon problem. Therefore, the NRAPs of Member States can have different quantitative objectives or targets (e.g., to reduce radon levels in different percentages of all the estimated dwellings exceeding the Reference Levels in the country), also considering the graded approach recommended for all radiation protection issues, including radon protection. Optimum indicators should be useful for any different target.

Indicators are not intended to compare the NRAPs of different Member States, but provide them with useful tools to evaluate the effectiveness of the actions included in the NRAP and to monitor the progress towards the targets.

The choice of indicators can also depend on the availability of national databases, e.g., databases collecting performed radon concentration measurements and remedial actions, or require the set-up of such databases.

### 6.1. The HERCA Activities on Indicators

On the basis of the above considerations, the HERCA has discussed within the Working Group on Natural Sources of Radiation (WG NAT) how to contribute to the identification of adequate indicators for NRAPs. It was decided that indicators of effectiveness and progress will be a major item of the Second HERCA workshop on NRAPs. The preparation of a questionnaire on indicators is ongoing in order to collect the national information from the HERCA members, which will be presented and discussed during the Second HERCA workshop on NRAPs.

Based on the experience of WG NAT members, the following relevant types of effectiveness indicators should be considered (and will be included in the questionnaire):*Indicators on surveys and radon concentration measurements*: e.g., the number of measured units (dwellings, etc.) and the percentage of administrative units with a (reliable) estimate of radon distribution and the population living there;*Indicators on exposure distribution and on the identification of units exceeding the national reference levels (RLs)*: e.g., the estimated number (and/or the percentage) of dwellings exceeding RLs, the number (and/or the percentage) of identified dwellings with radon concentration above RL;*Indicators on remedial actions in existing buildings and preventive measures in new ones*: e.g., the remediation rate over the RL (i.e., the number of remediated dwellings, compared with the estimated number of dwellings exceeding RL), the remediation rate below the RL (i.e., the number of remediated dwellings with initial Rn level below the RL, in case remedial actions also recommended below RL), the number (and/or the percentage) of new dwellings with preventive measures against radon;*Indicators on training on remedial/preventive actions and on available experts/services*: e.g., the number of periodic courses on radon remediation and prevention, and the number of qualified experts/services for radon remedial actions;*Indicators on public information and on public radon awareness*: e.g., the number of contacts to web sites and other social media containing information on radon protection;*Indicators on the overall impact*: e.g., the overall quantity of avoided exposure, the estimated number (and/or percentage) of avoided lung cancers due to the exposure reduction.

Moreover, the availability (or planned implementation) of national databases to collect information on radon measurements, remedial actions and other relevant data are also important aspects to be considered and will be included in the questionnaire.

Some of the indicators could also be specific for radon priority areas in order to take into account the requirements of the EU-BSS directive regarding such areas. For example, it would be important to evaluate the number (or percentage) of dwellings/workplaces estimated to exceed the RL in radon priority areas, compared with the total number of dwellings/workplaces exceeding RL in the whole country. In fact, in some cases—depending on the adopted criteria to define such areas, on the distribution of radon levels in such areas and on the population living therein compared with the rest of the country—the dwellings/workplaces exceeding RL in radon priority area can be comparable or significantly less than those in the rest of the country [21,28].

### 6.2. Example of Indicators Included in NRAPs of Four European Countries

Examples of the indicators used by some European countries (Ireland, France, Norway, Portugal) are reported below. These examples can illustrate some of the differences and similarities among the current choices of individual countries and will be a useful basis, together with the information collected from all the HERCA MS through the questionnaire described above, for the analysis and discussion on indicators that will be carried out in the second HERCA workshop on NRAP. However, the authors do not propose those examples as a reference or representative of all the possible useful uses of indicators.

#### 6.2.1. Ireland

The indicators used to assess the effectiveness of the Irish NRAP are made up of two types: leading and lagging indicators.

*(1) Leading indicators:* These give a real-time measure of progress towards reducing exposure. These indicators can be used as reliable evidence that the long-term objective will be achieved. The measurement frequency of indicators varies from annually to every five years. The Leading indicators that are measured in the Irish NRAP are:The number of radon tests in homes (measured annually);The rate of radon mitigation among those who have high radon levels (measured every 5 years);The number of radon mitigation training courses held in Ireland and the number attending (measured annually, or as courses are held);The number of courses held aimed at informing Building staff of how to install radon prevention measured and the number attending (measured annually, or as courses are held);The number of radon risk assessments carried out in workplaces using the “Business electronic Safety Management” known as BeSmart, which is hosted by the Health and Safety Authority in Ireland (measured annually);The rate of successful outcomes for those who carry out radon mitigation (measured every 5 years);The number of radon tests carried out in a dwelling that is linked to buying and selling of properties (measured annually);The number of website hits on a dedicated radon website (www.radon.ie, accessed on 7 February 2022).

It should be noted that there have been significant challenges associated with gathering some of these metrics. For example, it has not been possible to gather data for tests linked to buying and selling of properties, and some metrics require access to measurement data that is not always available from commercial measurement companies.

*(2) Lagging indicators:* These complement the leading indicators and provide information that may not be sufficiently timely to helpfully direct ongoing actions. The Lagging Indicators used in the Irish NRAP are:The population-weighted national average indoor radon concentration. This is used to estimate the number of lung cancers in Ireland that can be linked to radon (measured every 8 years) [29].The geographic weighted national average indoor radon concentration. This is used to assess the effectiveness of radon interventions, for example, the building regulation requirements (measured every 8 years) [30].Radon awareness levels (measured every 3 to 5 years).

It should be noted that, despite a reduction in the geographic weighted national average indoor radon concentration [30], a higher value of the annual number of radon-related lung cancers (from 300 to 350) was estimated compared with the previous estimate [29]. This is because of the large number of variables that affect the annual number of radon-related lung cancers. These variables include smoking rates, population age and population distribution.

#### 6.2.2. France

Measuring the health impact through a change in the number of radon-induced lung cancers can only be evaluated over the long term. Similarly, data on the average indoor radon concentration in buildings, reflecting the exposure of the population, is also only available on a long-term basis. For this reason, intermediate indicators allowing indirect evaluation of the reduction in exposure have been defined to evaluate the effectiveness of the national strategy implemented under the 4th French NRAP (2020–2024). They were chosen for their pertinence and the available data enabling them to be monitored.

These indicators aim to monitor the implementation of the Plan per sector: (i) general public, (ii) workplaces and (iii) buildings open to the public.

(i)
*Indicators for the general public sector*
Number of local radon information operations;Number of dwellings screened during local radon information operations;Perception of the radon risk among the French population.
(ii)
*Indicators for the workplaces sector*
Number of workplaces with a result above 300 Bq/m^3^ after concentration reduction works;Number of workers who receive radon exposure individual dosimetry monitoring;Number of workers who exceeded 20 mSv effective dose over 12 consecutive months;Number of radiation protection advisers trained for radon.
(iii)
*Indicators for the public buildings sector*
Number of buildings open to the public: screened, exceeding the reference level of 300 Bq/m^3^ and exceeding the threshold of 1000 Bq/m^3^;Number of buildings in which work has been carried out;Number of buildings in which additional measurements have been carried out as part of an assessment process.


#### 6.2.3. Norway

In Norway, the NRAP was first adopted by the government in 2009. It had two strategic goals: “Work towards reducing radon levels in all buildings and premises to below the stated limits, and, contribute to reducing radon exposure in Norway to as low as reasonably achievable.” In addition, separate sub-targets were set for the six sub-strategies, which dealt with land planning, new build, existing dwellings, communities exposed to especially serious radon problems, public buildings including schools and kindergartens and radon in the workplace.

In 2020, the NRAP was evaluated. However, it was not straightforward to measure the impact and effectiveness of the NRAP. Therefore, specific indicators were created for each of the six sub-targets. The purpose of these indicators was not to measure what is achieved but to measure the effect the achievements have had in society. A report containing the evaluation of the national strategy on radon 2009–2020 is available [31].

The indicators were based on surveys or by obtaining statistics from other sources, preferably with a zero measurement from before the NRAP first was published. They will be used to assess future achievement and are included in the proposal for the new NRAP, which is currently being under consideration and will be published in 2022.

Examples of considered indicators are the followings:*The percentage of homeowners who have measured radon*: Every three years, the DSA carries out a survey of how aware the public is of various radiation protection issues. Included in this is a question of whether you have made a radon measurement in your home or not. In addition to providing information on how many homes are measured, it is also a good measure of how aware the public is of radon. In the first years of the NRAP, the increase was good, but in recent years the increase has stopped (2008: 8%, 2012: 14%, 2014: 22%, 2017: 24%, 2020: 21%). This is noted as one of the major challenges for the next NRAP.*The number of existing homes being remediated*: A survey was carried out among the radon remediation companies. In collaboration with the Norwegian Radon Association, a questionnaire was distributed asking how many existing homes the individual companies had remediated in recent years. From the answers, the number of annual remediated homes could be calculated. Between 2016 and 2019, an average of 1500 homes had been remediated in Norway annually. From surveys, we know that approximately 150,000 homes have radon levels above 200 Bq/m^3^. By continuing this remediation rate, it would take a hundred years to remediate all homes above 200 Bq/m^3^, and the conclusion was that the remediation rate must increase and that this is a major challenge for the next NRAP.*The radon level in new buildings*: One target in the NRAP was that new buildings should have low radon concentrations. In 2010, legally binding limits on radon concentration for new buildings were introduced in the technical building regulations, in addition to mandatory preventive measures. To measure the effect of this regulation, two national surveys of radon in newly built homes were carried out in 2008 and 2016. Homes were randomly selected from the National Building Registry and invited to join the survey. The results showed a considerable reduction in radon concentrations after the implementation of new regulations. A statistically significant reduction was found for detached houses where the average radon concentration was almost halved from 76 to 40 Bq/m^3^ [32].*Mean radon exposure in homes*: Approximately every ten years, the DSA carries out surveys of indoor radon in dwellings. Dwellings or persons are randomly selected from national registers and are invited to join the survey. Mean radon exposure to the public or radon concentration in dwellings is a good indicator, but it is important to be aware that it is not necessarily easy to compare the results from different surveys if the surveys are not conducted with the exact same method every time.

#### 6.2.4. Portugal

To support the evaluation of the NRAP in Portugal, a set of metrics comprising two types of indicators were considered. Efficiency Indicators, which refer to the achievement of the measures within the stipulated timeframe and the Effectiveness Indicators, which are complementary indicators that provide evidence that the long-term objectives will be achieved (e.g., the number of tests carried out in homes and workplaces, the remediation rate in existing buildings, number of service providers recognised, number of hits on the website and on the potential map). While Efficiency Indicators are only associated with strict compliance with the defined timeline, Effectiveness Indicators are more complex and are closely linked to the specificity of the actions defined in the plan.

The following Effectiveness Indicators were defined for the Portuguese NRAP. They are related to (i) the population exposure to radon, (ii) the quality of the building stock, and (iii) the governance.

(i)Population exposure to radon

It assesses the contribution of the NRAP in reducing the occurrence of adverse effects on human health from prolonged exposure to radon.

*Health risks**:* Indicators that will assesses the risks to which the population is exposed (epidemiological study):Reduction in the annual radon concentration (%);Prevalence of neoplasms/lung cancer due to radon (No).

*Worker exposure:* Indicators that will assess the mechanisms of radon management in workplaces and the protection of workers:Exposed workers to >6 mSv (No);Protective measures (No);Workplaces tested (No);Workplaces remediated (No).

*Demographic structure of the population exposed to radon:* Indicators that will assess the age structure of the population, gender class and geographical distribution of the population exposed to radon:Age structure of the population (No and %);Population distribution by gender (No and %);Population distribution by prone areas (No and %).
(ii)Quality of the building stock

It assesses the contribution of the plan in improving the characteristics of the building stock (dwellings and workplaces) for protection against radon, both in the construction of new buildings (preventive measures) and in existing buildings (remedial or corrective measures).

*Buildings:* Indicators that will evaluate the distribution of the buildings, their age of construction and the existence of facilities (heating, insulation):Buildings by age of construction and materials used in construction (No);State of conservation of buildings (No).

*Constructive solutions:* Indicator that will evaluate the existing regulations and standards regarding constructive guidelines:Building regulations and standards (No).

*Housing stock costs:* Indicator that will evaluate the costs of building stock with preventive measures installed:Construction price per m^2^ (€).

*Energy Efficiency**:* Indicator that will evaluate the relation between the energy savings measures that are in place and the indoor air quality.

Buildings with energy savings measures and radon concentrations >300 Bq/m^3^ (No and %).

(iii)Governance

It assesses the level of articulation and capacity development of the entities involved in radon management.

*Institutional articulation:* Indicators that will evaluate how the existing institutional articulation allows the management of radon, defines the responsibilities, defines the competencies in the management of ionising radiation and if there are financing resources for the implementation of the plan.

Organisational structure;Human resources (No);Financial allocation/Costs (€);Financial support for testing and mitigation (€).

*Technical skills**:* Indicators that will assess the existing mechanisms for the technical capacity of stakeholders involved in radon management, namely professionals from public institutions.

Existence of technical documentation to support radon management (No);Training/awareness-raising actions for key players (No);Specialists in radon management (No);Specialists in radon mitigation (No).

*Accreditation of measurement and mitigation services**:* Indicators that will assess the levels of standardisation/accreditation of existing services providers and building materials certification.

Accredited/recognised services (No);Accredited/recognised companies (No);Certified materials (No).

*Raising awareness of society and stakeholders**:* Indicators that will evaluate how the plan contributes to disseminating information and raising awareness among the population and stakeholders.

Media communications (No);Targeted communications (No);Engagement actions (No);Radon stakeholder associations (No).

The NRAP is a 5-year plan; these indicators will be key to the review process and to support the development of the subsequent NRAP, identifying which strategies should continue and proposing new approaches that may be needed for the new 5-year cycle.

## 7. Conclusions

Protection from radon exposure is a challenge for radiation protection. The National Radon Action Plan is a necessary tool, required by international regulations, to properly deal with this challenge. However, there are no standards for developing, implementing and evaluating NRAPs. Moreover, taking into account the different situations among countries as regards radon exposure levels and other relevant factors, flexibility and graded approaches have to be considered in NRAPs.

Therefore, HERCA has carried out several workshops and other activities to support competent authorities in this difficult task. The ongoing activities are focused on effectiveness indicators which are necessary for evaluating the progress of the NRAPs, in order to allow a periodical review of the planned actions as also required by the European Directive 2013/59/Euratom and to evaluate the effectiveness in reaching the goal of significantly reducing the number of lung cancers attributable to radon exposure.

An agreed common set of adequate indicators could be a very useful tool to reach this important goal for public health.

## Figures and Tables

**Figure 1 ijerph-19-04114-f001:**
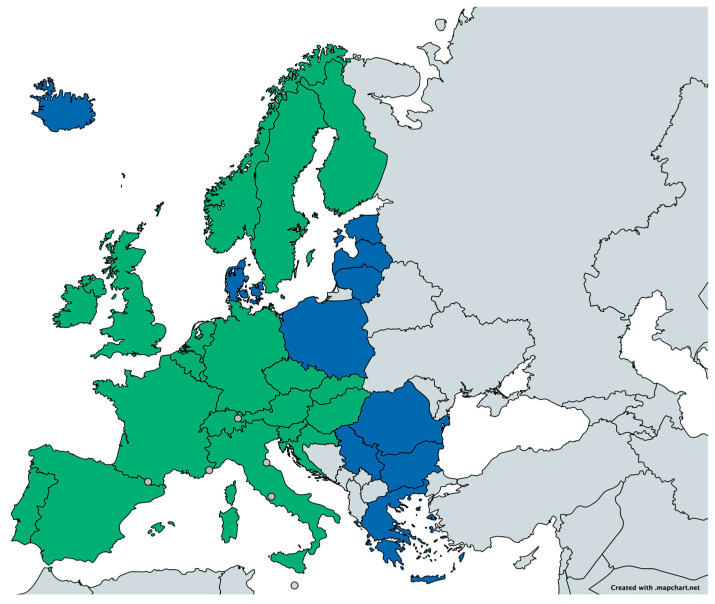
Map of the 20 HERCA member countries (in green) that provided information on the implementation of EU-BSS on radon in workplaces. Other HERCA member countries are in blue.

**Figure 2 ijerph-19-04114-f002:**
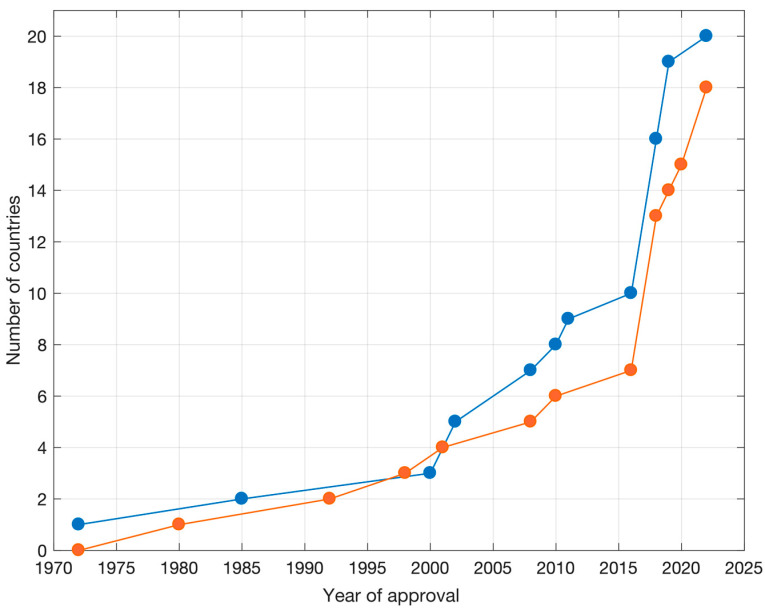
Number of countries vs. year of approval of regulation on radon protection in: workplaces (blue); dwellings and other indoor environments (existing or newly built) (orange).

**Figure 3 ijerph-19-04114-f003:**
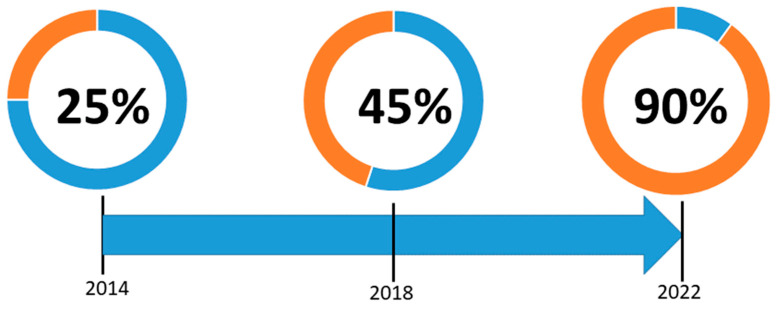
Time evolution in the percentage of HERCA countries having NRAPs in place. At 2014: year of approval of directive 2013/59/Euratom (EU-BSS) and first HERCA Workshop on NRAPs; 2018: deadline for the transposition of the EU-BSS; and 2022: by February, when the last review was conducted.

**Figure 4 ijerph-19-04114-f004:**
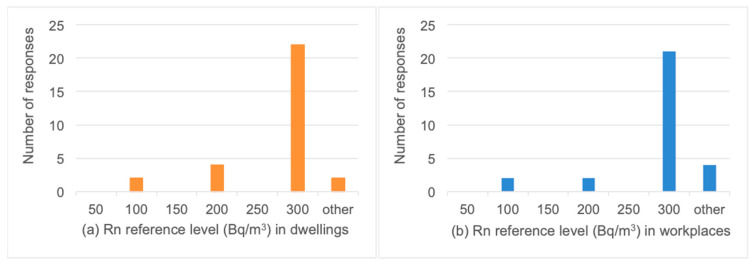
Reference levels for radon concentration in: (**a**) dwellings (orange), (**b**) workplaces (blue).

## Data Availability

Not applicable.

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
