# Peer review of "National Radon Action Plans in Europe and Need of Effectiveness Indicators: An Overview of HERCA Activities"

_ijerph, 2022, doi:10.3390/ijerph19074114_

Round 1
Reviewer 1 Report
The article is very good and valuable, it is well organized and clearly written. It deals with an important topis at the moment, i.e. protection of the population and workers against exposure to radon.
I found only a few misprints in the text. The abbreviation WG NAT is written in different ways: WGNAT, WG NAT, WG-NAT. I suggest to use the homogeneous form. There is also a dot missing in the list on page 13, line 560.
Reviewer 2 Report
Dear Author/s,
The manuscript has been well-written and could answer all concerns regarding the research propose.
A minor correction is required:
- in abstract please use full name of WHO organization as before it was not pre-defiend the "WHO"
- some places BSS and other EU-BSS were used, please use a uniform style.
Thank you
Reviewer 3 Report
In the attached pdf file, the typo errors and others of that type are indicated.
Abstract:
In this review I will try to be brief, I will not make the initial description of the article. The Abstract is very good.
- Introduction
The Introduction presents the background of measures against exposure to radon in interiors and workplaces. In addition, the actions and measures recommended by European and international organizations are exposed.
The Introduction concludes by stating that the issue of indicators is presented and discussed, reporting both the HERCA on-going activities on this issue and examples from the NRAP of some European countries.
- Evolution of the legal framework on protection against radon in Europe and NRAPs
There are some typo and syntax errors in this section.
The texts that I consider most relevant in this section give quality to the entire article, as they indicate the route to be followed by European nations in terms of measures to reduce the risk of radon inhalation.
“Radon exposure being a public health problem, the resources and efforts dedicated to the NRAP need to be commensurate to other health-related policies in the country.”
“Emphasis on optimization below the reference level, is also needed. This is also in line with the principle of optimisation and the graded approach for radioprotection introduced by ICRP in Publ.103, which recommended “reference level” (RL) to be used as a tool towards optimization, replacing the previous “action level” (AL).”
Lines 185-187: This text has to be rewritten. It is difficult to understand as it is now:
“Optimisation is with priority for initial levels above reference level (RL), but also for levels lower than the RL, whereas the previous AL tool required to consider remedial actions only for initial levels above AL.”
- First HERCA workshop on NRAP (2014)
There are some typo and syntax errors in this section.
The texts that I consider most relevant in this section is given below:
"The workshop was initiated in response to the EU-BSS directive requirement for Member States to define and adopt an NRAP for reducing radon exposure.
The importance of intensive multi-level collaboration was highlighted, and it was considered efficient having one authority that coordinates the radon reduction activities and follows-up the NRAP. It was also agreed that radon risk communication should be an important aspect of a NRAP. Information on radon and related health risk should be given to different groups of people including homeowners, landlords, employers, solicitors, estate agents, building professionals, architects, radon remediators, officers in local and national government and family doctors.
From the discussions during the workshop, it became clear that a NRAP should not only aim to reduce the high radon levels. It should also aim to reduce the average radon concentrations in the housing stock and other public access buildings and premises, thus reducing the overall lung cancer risk."
- The HERCA workshop on radon in workplaces (2015)
There are some typo and syntax errors in this section.
The inhalation equivalent dose values for radon mentioned in this text are quite high. This information in itself helps the reader to get an idea of the problem in European countries, where winter intervenes significantly in the tendency to close houses. The texts that I consider most relevant in this section is given below:
“Different viewpoints were shared on how to understand the specific requirements dealing with radon exposure in workplaces.
The issues raised were mainly focused on:
- The justification of actions to reduce radon exposures in the establishment and implementation of the National Radon Action Plan, being under the responsibilities of the Government and regulators;
- Identification of workplaces, radon measurements and control, the employer’s and/or the undertaking’s responsibilities depending on workplace type;
- The use of the reference level concept and how it supports but should not be confused with dose limitation.
The EU-BSS directive requiring management of the radon risk by both the concentration of radon in air and the effective dose, the availability and the use of international guidelines, and associated tools, to calculate annual effective doses, was identified as a key point. In case of doses exceeding 6 mSv per year, it was recommended to apply the occupational exposure requirements related to optimisation, to the radiological surveillance of workplaces (adapted to radon exposure), to workers’ information and, in some cases, to individual monitoring.”
- Second HERCA Workshop on NRAPs (2022) and the pre-workshop event (2021)
The texts that I consider most relevant in this section is given below:
“The follow up NRAP workshop was planned in autumn 2020, but due to world pandemic conditions was postponed twice, and is currently planned for June 2022.
However, to meet the significant international interest in NRAP issues, an international event as an introduction to the second HERCA NRAP workshop was organized as an online event in March 2021.
Based on the presentations, the panel discussion and the results of a received survey, key messages of this pre-workshop event were as follows:
Radon prevention in the new buildings, the approach to radon in workplaces and related risk communication, and raising awareness and engagement level are examples of core issues where sharing of lessons learned, best practices and challenges is of high importance;
Evaluation of taken measures is necessary for an assessment of their effectiveness, but also for further actions and the decision-making on recommended versus legally binding, compulsory approaches.”
- Effectiveness indicators
The texts that I consider most relevant in this section is given below:
“It is worth noting that considerable differences exist between countries in the approaches used to address the radon risk. This is understandable given diverse climates, geologies and building practices that exist between countries. Therefore, the NRAPs of Member States can have different quantitative objectives or targets, also considering the graded approach recommended for all radiation protection issues including radon protection. Optimum indicators should be useful for any different target.
Indicators are not intended to compare NRAP of different Member State, but to provide them useful tools to evaluate the effectiveness of the actions included in the NRAP and to monitor the progress towards the targets.”
Lines 416-420: This sentence is too long and has too many subordinate clauses.
6.1. The HERCA activities on indicators
The texts that I consider most relevant in this section is given bellow:
“Based on the experience of WG-NAT members, the following relevant types of effectiveness indicators should be considered (and will be included in the questionnaire):
- Indicators on surveys and radon concentration measurements
- Indicators on exposure distribution and on the identification of units exceeding the national reference levels (RLs)
- Indicators on remedial actions in existing buildings and preventive measures in new ones
- Indicators on training on remedial/preventive actions and on available experts/services
- Indicators on public information and on public radon awareness
- Indicators on the overall impact.
Some of the indicators could also be specific for radon priority areas, in order to take into account, the requirements of the EU-BSS directive regarding such areas.”
6.2. Example of indicators included in NRAPs of four European countries
The examples in section 6.2 are very different and very complex to analyze. They were probably written by each of the protagonists and they understand them very well, but the reader does not understand them. Above all, the reader cannot know which of the examples is more suited to the situation in which he is potentially interested and more useful to assimilate.
I suggest significantly reducing that section, making diagrams or tables of each case instead of giving extensive texts. In any case, these long texts may be the content of annexes or the new schemes may refer to the publications that support them.
- Conclusions
Perhaps it would be more useful to make a summary of the situation and the measures to be taken by HERCA, instead of this very brief text.

Reviewer 4 Report
This work summarizes and outlines the main results and conclusions obtained in the process of implementing National Radon Action Plans (NRAP) in several European countries. A description of the historical evolution of protection against radon as well as the accompanying EU legal framework is provided. The main results obtained implementing NRAP's are explained through a summary of the conclusions and results obtained in two workshops organized by the Heads of European Radiological Protection Competent Authorities (HERCA) in 2014-2015 and 2021-2022. Finally, several examples of indicators used to monitor the progress obtained implementing the NRAPs in several European countries are listed.
This is an interesting and clearly written work that provides an interesting review and summary of the main actions that have been implemented in Europe with the objective to protect the population against the risks related to radon exposure. I do believe that it should be published, although I have suggestions for the authors:
1. The quality of figure 1 is low, could the authors provider a higher quality version of this image?
2. If I understand properly figure 2, I would say that it is confusing. I understand that it is not intended to be scaled, but it actually is really out of scale. If the first point represents a 21%, the second point a 36% and the last one 100%, the arrow curvature should be the opposite. The higher increase is obtained between 2018 and 2022, not between 2014 and 2018.
3. In line 179 the authors say "..., most radon attributable cancers occur at concentrations below $100 Bq/m^3$". It would be advisable to provide some reference to support this claim?
I do also have some minor additional suggestions:
4. In line 208 there are two consecutive dots.
5. In line 391, "use" should be "used".
6. In section 6.2 Ireland, France, Norway and Portugal are chosen as examples. Why have the authors selected these countries? Do they represent alternative and representative examples?
In summary, I do think that this is an interesting and useful work, and I do recommend its publication in the International Journal of Environmental Research and Public Health, provided that the authors deal with my previous suggestions.
